# Fundamental limits on inferring epidemic resurgence in real time using effective reproduction numbers

Kris V. Parag [1]*, Christl A. Donnelly [1,2]

**1** MRC Centre for Global Infectious Disease Analysis, Imperial College London, London, United Kingdom,
**2** Department of Statistics, University of Oxford, Oxford, United Kingdom

* k.parag@imperial.ac.uk

**Data Availability Statement:** All data and code used to generate the analyses and figures of this work are freely available at: https://github.com/kpzoo/resurgence-detection.

## Abstract

We find that epidemic resurgence, defined as an upswing in the effective reproduction number ($R$) of the contagion from subcritical to supercritical values, is fundamentally difficult to detect in real time. Inherent latencies in pathogen transmission, coupled with smaller and intrinsically noisier case incidence across periods of subcritical spread, mean that resurgence cannot be reliably detected without significant delays of the order of the generation time of the disease, even when case reporting is perfect. In contrast, epidemic suppression (where $R$ falls from supercritical to subcritical values) may be ascertained 5–10 times faster due to the naturally larger incidence at which control actions are generally applied. We prove that these innate limits on detecting resurgence only worsen when spatial or demographic heterogeneities are incorporated. Consequently, we argue that resurgence is more effectively handled proactively, potentially at the expense of false alarms. Timely responses to recrudescent infections or emerging variants of concern are more likely to be possible when policy is informed by a greater quality and diversity of surveillance data than by further optimisation of the statistical models used to process routine outbreak data.

## Author summary

The timely detection of epidemic resurgence (i.e., upcoming waves of infected cases) is crucial for informing public health policy, providing valuable signals for implementing interventions and identifying emerging pathogenic variants or important population-level behavioural shifts. Increases in epidemic transmissibility, parametrised by the time-varying reproduction number, $R$, commonly signify resurgence. While many studies have improved computational methods for inferring $R$ from case data, to enhance real-time resurgence detection, few have examined what limits, if any, fundamentally restrict our ability to perform this inference. We apply optimal Bayesian detection algorithms and sensitivity tests and discover that resurgent (upward) $R$-changes are intrinsically more difficult to detect than equivalent downward changes indicating control. This asymmetry derives from the often lower and stochastically noisier case numbers that associate with resurgence, and induces detection delays on the order of the disease generation time. We

**Funding:** KVP and CAD acknowledge funding from the MRC Centre for Global Infectious Disease Analysis (reference MR/R015600/1), jointly funded by the UK Medical Research Council (MRC) and the UK Foreign, Commonwealth & Development Office (FCDO), under the MRC/FCDO Concordat agreement and is also part of the EDCTP2 programme supported by the European Union. CAD thanks the UK National Institute for Health Research Health Protection Research Unit (NIHR HPRU) in Emerging and Zoonotic Infections in partnership with Public Health England (PHE) for funding (grant HPRU200907). The funders had no role in study design, data collection and analysis, decision to publish, or manuscript preparation.

**Competing interests:** The authors have declared that no competing interests exist.

prove these delays only worsen if spatial or demographic differences in transmissibility are modelled. As these fundamental limits exist even if case data are perfect, we conclude that designing integrated surveillance systems that fuse potentially timelier data sources (e.g., wastewater) may be more important than improving $R$-estimation methodology and deduce that there may be merit (subject to false alarm costs) in conservative resurgence response initiatives.

## Introduction

Real-time estimates of the transmissibility of an infectious disease [1,2] are crucial for informed outbreak responses. Timely detection of salient changes in the effective reproduction number ($R$) of the disease of interest, which measures the average number of secondary cases likely caused by a typical primary case, can provide important evidence for policymaking and public communication [3,4], as well as improve forecasts of disease burden [5] (e.g., hospitalisations and deaths). Two critical changes of interest are *resurgence* and *control*. Resurgence, which we define as an increase from subcritical ($R \leq 1$) to supercritical ($R > 1$) transmissibility, can warn of imminent waves of infections, signify the emergence of pathogenic variants of concern and signal important shifts in the behavioural patterns of population [6,7]. Alternatively, control (or suppression) describes a switch from supercritical to subcritical spread and can indicate the effectiveness of interventions and the impact of depleting susceptibility (including that due to vaccine-induced immunity) [8,9].

Identifying these transmissibility changes in real time, however, is an enduring challenge for statistical modelling and surveillance planning. Inferring a transition in $R$ from stochastic time series of incident cases necessitates assumptions about the differences among meaningful variations (*signal*) and random fluctuations (*noise*) [10–12]. Modern approaches to epidemic modelling and monitoring aim to maximise this signal-to-noise ratio either by enhancing noise filtering and bias correction methods [13–15], or by amplifying signal fidelity through improving surveillance quality and diversity [16–18]. While both approaches have substantially advanced the field, there have been few attempts to explore what, if any, *fundamental limits* exist on the timely detection of these changes. Such limits can provide key benchmarks for assessing the effectiveness of modelling or data collection and deepen our understanding of what can and cannot be achieved by real-time outbreak response programmes, ensuring that model outputs are not overinterpreted and redirecting surveillance resources more efficiently [19–21].

While studies are examining intrinsic bounds on epidemic monitoring and forecasting [22–25], works on transmissibility have mostly probed how extrinsic surveillance biases might cause $R$ misestimation [14,26–28]. Here we address these gaps in the literature by characterising and exposing fundamental limits on detecting resurgence and control, from a perfectly ascertained incidence time series, using effective reproduction numbers. This presents new and useful insights into the best real-time performance possible and blueprints for how outbreak preparedness might be improved. We analyse a predominant, flexible real-time epidemic model [1,2] and discover stark asymmetries in our intrinsic ability to detect resurgence and control, emerging from the noisier, low-incidence data underlying possible resurgence events. While epidemic control or suppression change-points are inferred robustly and rapidly, the data bottleneck caused by subcritical spread forces inherent delays (potentially 5–10 times that for control and on the order of the mean disease generation time) that inhibit real-time resurgence estimation.

We show that these innate constraints on resurgence detection worsen with smaller epidemic size, steepness of the upswing in $R$ and spatial or demographic heterogeneities. Given these limitations to timely outbreak analysis, which exist despite perfect case reporting and the use of optimal Bayesian detection algorithms [15,29], we argue that methodological improvements to existing models for analysing epidemic curves (e.g., cases, hospitalisations or deaths) are less important than designing enhanced and integrated surveillance systems [30,31]. Such systems, which might fuse multiple data streams including novel ones (e.g., wastewater [32]) to triangulate possible resurgences, could minimize some of these fundamental bottlenecks. We conclude that early responses to suspected resurging epidemics, at the expense of false alarms, might be justified in many settings, both from our analysis and the consensus that lags in implementing interventions can translate into severely elevated epidemic burden [33–36]. While such decisions must, ultimately, be weighed against the cost of those interventions, the bottlenecks we expose, hopefully, bolster the evidence base for decision-making. Using theory and simulation, we explore and elucidate these conclusions in the next section.

## Results

### Epidemic resurgence is statistically more difficult to infer than control

We first provide intuition for why resurgence and control might present asymmetric difficulties when inferring transmissibility in real time. We consider an epidemic modelled via a renewal branching process [37] over times (usually in days) $1 \leq s \leq t$. Such models have been widely applied to infer the transmissibility of many diseases including COVID-19, pandemic influenza and Ebola virus disease. Renewal models postulate that the incidence of new cases at time $s$, denoted $I_s$, depends on the effective reproduction number, $R_s$, and the past incidence, $I_1^{s-1}$ as in **Eq (1)** [2]. Here $I_a^b$ means the set or time series $\{I_a, I_{a+1}, \ldots, I_b\}$ and $\equiv$ indicates equality in distribution.

$$P(I_s | R_s, I_1^{s-1}) \equiv \text{Pois}(R_s \Lambda_s), \qquad \Lambda_s = \sum_{u=1}^{s-1} w_u I_{s-u}. \qquad (1)$$

In **Eq (1)**, Pois represents Poisson noise and $\Lambda_s$ is the total infectiousness, which summarises the weighted influence of past infections. The set of weights $w_u$ for all $u$ define the generation time distribution of the infectious disease with $\sum_{u=1}^{\infty} w_u = 1$ [38]. We assume that all $w_u$ are known. If this distribution changes across the epidemic [39], we can recompute the $\Lambda_s$ terms after that change to model its effects. Applying Bayesian inference techniques (see **Methods** for all derivations) [2,40] under the assumption that transmissibility is constant over a past window of size $m$ days, $\tau(s) = \{s, s-1, \ldots, s-m+1\}$, we obtain the gamma (Gam) posterior distribution given the incidence data $P(R_s | I_1^s) \approx P(R_s | I_{s-m+1}^s) \equiv \text{Gam}(a + i_{\tau(s)}, (c + \lambda_{\tau(s)})^{-1})$, with sums across the window of $i_{\tau(s)} = \sum_{u \in \tau(s)} I_u$ and $\lambda_{\tau(s)} = \sum_{u \in \tau(s)} \Lambda_u$.

Here $(a, c)$ are prior distribution ($P(R_s)$) parameters, which are set so the prior mean of $R_s$ is above 1 but uninformative. This maximises sensitivity to resurgence since the model, in the absence of data, favours $E[R_s] > 1$. The approximations above and later emerge from the window assumption and underpin popular real-time $R$-inference methods [2,41]. Using this renewal formulation, we define the relative change in the epidemic size as $\Delta \lambda_{\tau(s)} = \frac{i_{\tau(s)} - \lambda_{\tau(s)}}{\lambda_{\tau(s)}}$. This measures the perturbation to the past incidence (summarised by $\lambda_{\tau(s)}$) that the most recently observed incidence, $i_{\tau(s)}$, causes over $\tau(s)$. Normalising by $\lambda_{\tau(s)}$ is sensible as the posterior mean estimate of $R_s$ is roughly $\frac{i_{\tau(s)}}{\lambda_{\tau(s)}}$, so $\Delta \lambda_{\tau(s)}$ approximates $R_s - 1$.

This posterior distribution only uses data up until time $s$ and defines our real-time estimate of $R$ at that time. We can analyse its properties (and related likelihood function $P(I_{s-m+1}^s | R_s)$) to

obtain the Fisher information (FI) on the left side of **Eq (2)**. We derive this expression in the **Methods**. This FI captures how informative $I_1^s$ is (here approximated by $I_{s-m+1}^s$) for inferring $R_s$, with its inverse defining the smallest asymptotic variance of any $R_s$ estimate [10,42]. Larger FI implies better statistical precision.

$$\text{FI}[R_s] = \frac{\lambda_{\tau(s)}}{R_s}, \qquad P(R_s > 1 | I_1^s) = \sum_{j=0}^{a-1+i_{\tau(s)}} \frac{(c + \lambda_{\tau(s)})^j}{j!} e^{-(c + \lambda_{\tau(s)})}. \qquad (2)$$

As resurgence will likely follow low-incidence periods, we might expect $\lambda_{\tau(s)}$ to be small, while $R_s$ rises. This effect will reduce the FI in **Eq (2)**, making these changes harder to detect. In contrast, the impact of interventions will be easier to infer since these are often applied when cases are larger (so $\lambda_{\tau(s)}$ will be big) and reduce $R_s$. This observation applies for any $\tau(s)$ and is fundamental as it delimits the best estimator performance under our renewal model (Cramer-Rao bound) [43].

We expand on this intuition, using the $R$ posterior distribution to derive (see **Methods**) the real-time resurgence probability $P(R_s > 1 | I_1^s) \approx \int_1^\infty P(R_s | I_{s-m+1}^s) dR_s$, as on the right side of **Eq (2)**. We plot its implications in **Fig 1**, corroborating our intuition. In panel A we find that larger past epidemic sizes ($\lambda_{\tau(s)}$) improve our ability to detect transmissibility shifts from fluctuations in incidence (the posterior distributions for $R_s$ overlap less). Panel B bolsters this idea, showing that when $\lambda_{\tau(s)}$ is smaller (as is likely before resurgence) we need to observe larger relative epidemic size changes ($\Delta\lambda_{\tau(s)}$) for some increase in $P(R_s > 1 | I_1^s)$ than for an equivalent decrease when aiming to detect control (where $\lambda_{\tau(s)}$ will often be larger). This detection asymmetry holds for arbitrary window sizes and indicates that data bottlenecks translate into real-time detection delays. We assess the magnitude of these delays next.

## Fundamental delays on detecting resurgence but not control

The intrinsic asymmetry in sensitivity to upward versus downward shifts in $R$ (see **Fig 1**) implies that it is not equally simple to infer resurgence and control from incident cases. We investigate ramifications of this observation by comparing our real-time $R_s$-estimates to ones exploiting all the future incidence information available. We no longer consider window-based approximations (which we only use to extract analytic insights) but instead apply formal real-time Bayesian inference and detection algorithms [29]. We investigate two foundational posterior distributions, the *filtered*, $p_s$, and *smoothed*, $q_s$, distributions, defined as in **Eq (3)**.

$$p_s = P(R_s | I_1^s), \qquad q_s = P(R_s | I_1^t), \qquad D(p_s | q_s) = \int_0^\infty p_s \log \frac{p_s}{q_s} dR_s. \qquad (3)$$

Here $p_s$ considers all information until time $s$ and captures changes in $R_s$ from $I_1^s$ in real time. Estimates of $R_s$ using this posterior distribution minimise the mean squared error (MSE) given $I_1^s$. In contrast, $q_s$ extracts all the information from the full incidence curve $I_1^t$, providing the minimum MSE $R_s$-estimate given $I_1^t$ [29]. This smoother MSE is never larger and may be substantially smaller than the filtered MSE due to its use of additional information (i.e., $I_{s+1}^t$) [29,44]. The differential between $p_s$ and $q_s$, summarised via the Kullback-Liebler divergence, D $(p_s | q_s)$, measures the value of this additional 'future' information.

Bayesian filtering and smoothing are central formalisms across engineering, where real-time inference and detection problems are common [29,45]. We compute formulae from **Eq (3)** via the *EpiFilter* package (see [15,28]), which uses optimal forward-backward algorithms, improves on the window-based approach of the last section and maximises the signal-to-noise ratio in $R$-estimation. We further obtain filtered and smoothed probabilities of resurgence as

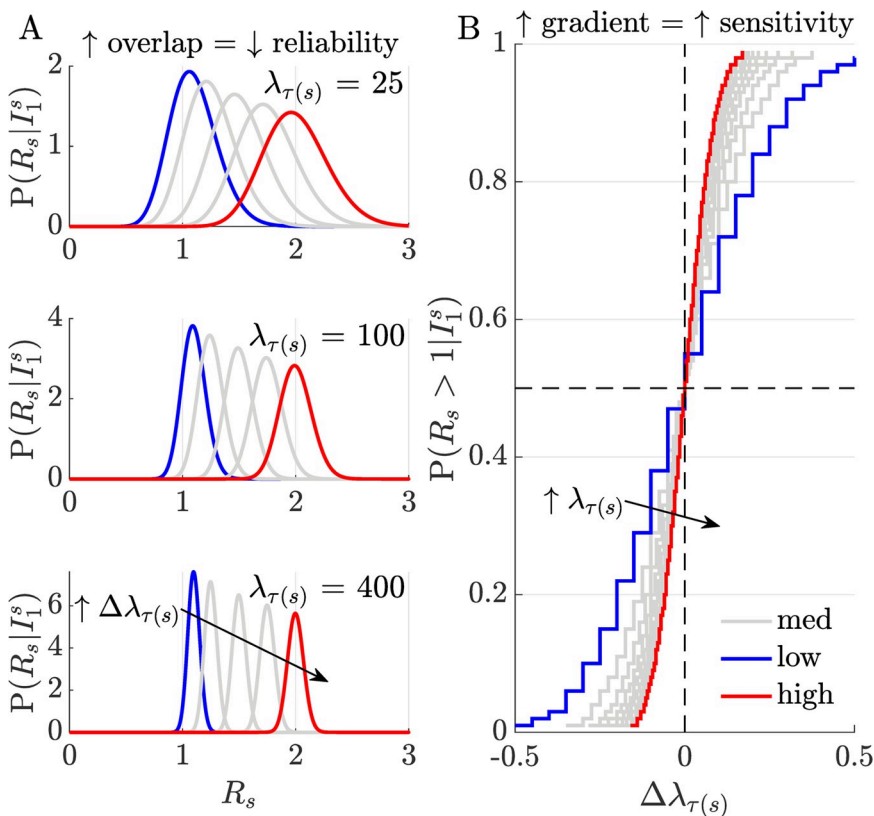

**Fig 1. Relative sensitivity to perturbations in incidence.** Panel A plots posterior real-time distributions for time-varying reproduction numbers $R_s$, given incidence data $I_1^s$, at different relative incidence perturbations, $\Delta\lambda_{\tau(s)} = \frac{i_{\tau(s)} - \lambda_{\tau(s)}}{\lambda_{\tau(s)}}$, (increasing from blue to red). Here $\tau(s)$ represents some arbitrary window size used in computation (see **Eq (2)**). The degree of distribution separation and hence our ability to uncover meaningful incidence fluctuations from noise, improves with the current epidemic size, $\lambda_{\tau(s)}$ (i.e., as this increases from 25–400 overlap among the distributions decreases). Panel B shows how this sensitivity modulates our capacity to infer resurgence ($P(R_s > 1|I_1^s)$) and control ($P(R_s \leq 1|I_1^s) = 1 - P(R_s > 1|I_1^s)$). If epidemic size is smaller, larger relative incidence perturbations are required to detect the same change in $R_s$ (curves have steeper gradient as we traverse from blue to red). Resurgence (likely closer to the blue line in the top right quadrant) is appreciably and innately harder to detect than control (likely closer to the red line, in the bottom left quadrant).

$P(R_s > 1|I_1^s) = \int_1^\infty p_s dR_s$ and $P(R_s > 1|I_1^t) = \int_1^\infty q_s dR_s$. The probability that the epidemic is controlled (i.e., $R \leq 1$) is the complement of these expressions. Our main results, which average the above quantities over many simulated Ebola virus and COVID-19 epidemics, are given in **Fig 2** and **Fig A** in the **S1 Appendix**, respectively. The simulated incidence curves are also provided in **Figs B-C** in the **S1 Appendix** and illustrate the expected differences in case numbers associated with both upward and downward shifts in $R$. We uncover striking differences in the intrinsic ability to infer resurgence versus control in real time.

Upward change-points are significantly harder to detect both in terms of accuracy and timing. Discrepancies between $p_s$- and $q_s$-based estimates (the latter benchmark the best realisable performance) are appreciably larger for resurgence than control. While decreases in $R$ can be pinpointed reliably, increases seem fundamentally more difficult to detect. These limits appear to exacerbate with the steepness of the $R$ upswing. We confirm these trends with a detailed simulation study across five infectious diseases in **Fig 3**. There we alter the steepness, $\theta$, of transmissibility changes and map delays in detecting resurgence and control as a function of

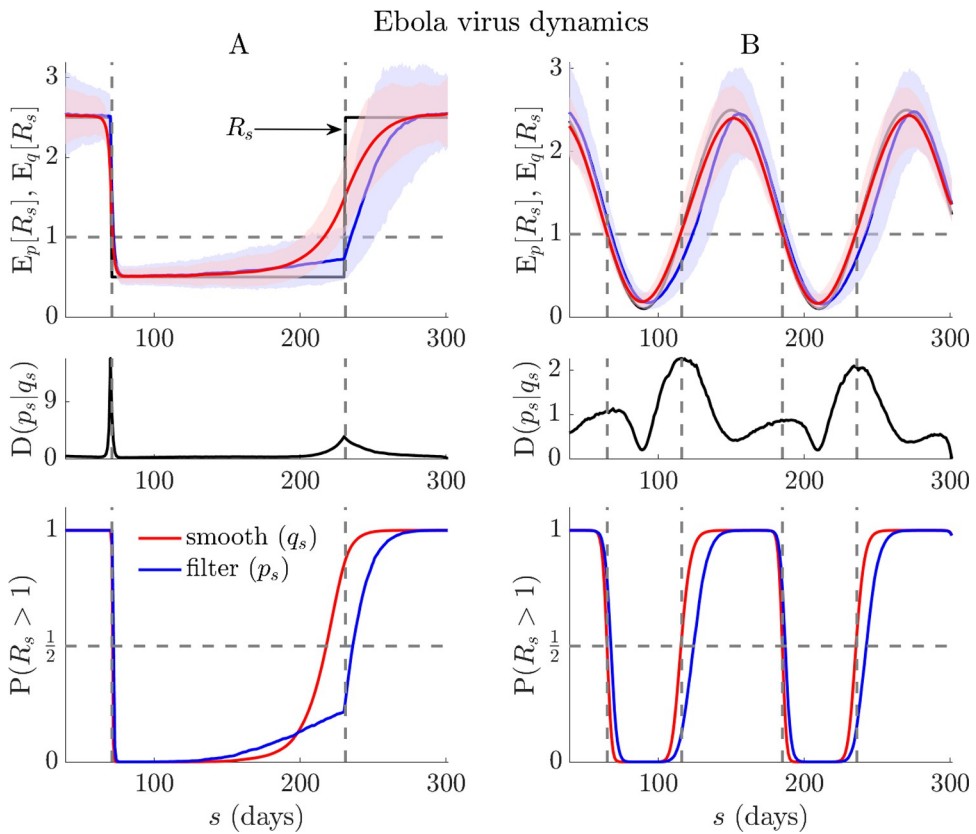

**Fig 2. Resurgence and control dynamics of Ebola virus.** Using renewal models with the generation time from [46], we simulate 1000 realisations of Ebola virus epidemics ($t = 300$) with step (A panels) and seasonally (B panels) changing transmissibility (true $R_s$ in black). Top panels show posterior mean estimates from the filtered ($E_p[R_s]$, blue) and smoothed ($E_q[R_s]$, red) distributions from every realisation (computed using *EpiFilter* [15]). Middle panels average the Kullback-Liebler divergences from those simulations, $D(p_s|q_s)$, and bottom panels display the overall filtered ($P(R_s > 1|I_1^s)$, blue), and smoothed ($P(R_s > 1|I_1^t)$, red) probabilities of resurgence. We find fundamental and striking delays in detecting resurgence, often an order of magnitude longer than those for detecting control or suppression in transmission (see lags between red and blue curves in all relevant panels). Note that the initial rise in $P(R_s > 1|I_1^s)$ of panel A, which precedes the transition in $R_s$, is due to the influence of the prior distribution (which has a mean above 1) in a period with very few cases. We present the incidence curves that underlie the simulations here in **Fig C** in the **S1 Appendix**.

the difference in the first time that $p_s$- and $q_s$-based probabilities cross 0.5 ($\Delta t_{50}$) and 0.95 ($\Delta t_{95}$), normalised by the mean generation time of the disease. We find that lags in detecting resurgence can be at least 5–10 times longer than for detecting control and are of the order of the average intrinsic generation time of the disease.

## Fundamental delays worsen with spatial or demographic heterogeneities

In previous sections we demonstrated that sensitivity to changes in $R$ is asymmetric, and that intrinsic, restrictive limits exist on detecting resurgence in real time, which do not equally inhibit detecting control. While those conclusions apply generally (e.g., across diseases), they do not consider the influence of spatial or demographic heterogeneity. We examine this complexity through a simple but realistic generalisation of the renewal model. Often $R$-estimates can be computed at small scales (e.g., at the municipality level) via local incidence or more coarsely (e.g., countrywide), using aggregated case counts [3,13]. We can relate these differing scales with the weighted mean in **Eq (4)**, where the overall (coarse) $R$ at time $s$, $\bar{R}_s$, is a convex

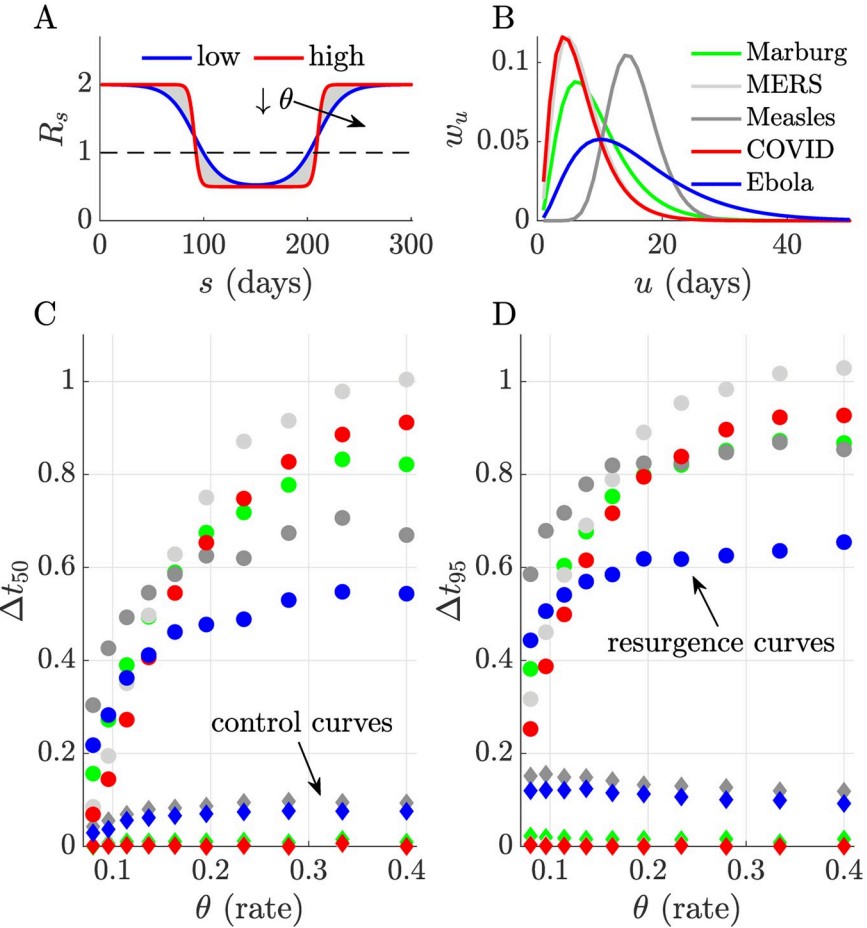

**Fig 3. Delays in detecting upward and downward changes in *R*.** We characterise the discrepancies between detecting resurgence and control against the steepness or rate, $\theta$, of changes in transmissibility ($R_s$), which we model using logistic functions (panel A, steepness increases from blue to red). We compare differences in the probability of detecting resurgence ($P(R_s>1)$) or control ($P(R_s\leq1)$ under filtered and smoothed estimates (see main text) first crossing thresholds of 0.5 ($\Delta t_{50}$) and 0.95 ($\Delta t_{95}$) for five infectious diseases (panel B plots their assumed generation time distributions from [2,46,47]). We simulate 1000 epidemics from each disease using renewal models and estimate $R_s$ with *EpiFilter* [15]. Panels C and D (here colours match panel B, $\Delta t$ is normalised by the mean generation times of the diseases) show that delays in detecting resurgence (dots with colours indicating the disease) are at least 5–10 times longer than for detecting control (diamonds with equivalent colours). Our ability to infer even symmetrical transmissibility changes is fundamentally asymmetric, largely due to the differences in case incidence at which those changes usually tend to occur.

sum of finer-scale *R* contributions from each group ($R_s[j]$ for the $j^{\text{th}}$ of $p$ groups) weighted by the epidemic size of that group (as in **Eq (2)** we use windows $\tau(s)$, of some size $m$, to derive analytic insight).

$$\bar{R}_s = \sum_{j=1}^{p} R_s[j]\alpha_j, \qquad \alpha_j = \frac{\lambda_{\tau(s)}[j]}{\sum_{k=1}^{p} \lambda_{\tau(s)}[k]}. \qquad (4)$$

Our choice of groupings is arbitrary and can equally model demographic heterogeneities (e.g., age-specific transmission), where we want to understand how dynamics within the sub-groups influence overall spread [7]. Our aim is to ascertain how grouping, which often occurs naturally due to data constraints or a need to succinctly describe the infectious dynamics over a country to aid policymaking or public communication [48], affects resurgence detection. **Eq**

(4) implies that $\bar{R}_s - 1 = \sum_{j=1}^{p} (R_s[j] - 1)\alpha_j$. Since resurgence will likely first occur within some specific (maybe high risk) group and then propagate to other groups [7], this expression suggests that an initial signal (e.g., if some $R_s[j] > 1$) could be masked by non-resurging groups (which are, from this perspective, contributing background noise).

As the epidemic size in a resurging group will likely be smaller than those of groups with past epidemics that are now being stabilised or controlled, this exacerbates the sensitivity bounds explored earlier via **Eq (2)**. We can verify this further loss of sensitivity by examining how the overall posterior distribution depends on those of the $p$ component groups as follows, with ⊛ as a repeated convolution operation and $\Omega_j$ as some generic posterior distribution for the $j^{\text{th}}$ group.

$$R_s[j] \sim \Omega_j(i_{\tau(s)}[j], \lambda_{\tau(s)}[j]), \qquad \bar{R}_s \sim \circledast_{j=1}^{p} \alpha_j \Omega_j. \qquad (5)$$

While **Eq (5)** holds generally, we assume gamma posterior distributions, leading to statistics analogous to **Eq (2)**. We plot these sensitivity results at $p = 2$ and 3 in **Fig 4**, where group 1 features resurgence and other groups either contain stable or falling incidence. We find that as $p$ grows (and additional distributions convolve to generate $\bar{R}_s$) we lose sensitivity (posterior

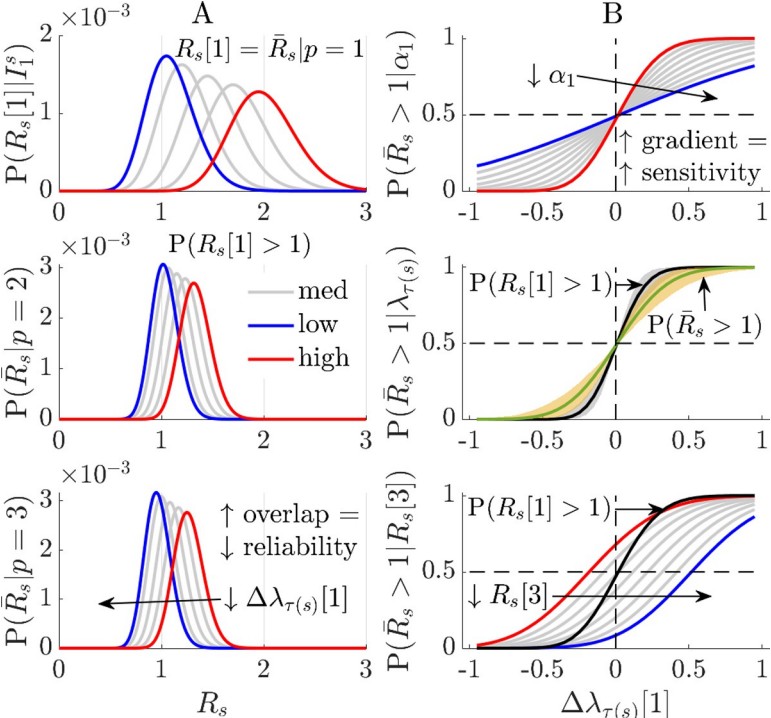

**Fig 4. Influence of heterogeneities in transmission.** We investigate how differences in transmissibility among groups (e.g., due to demographic or spatial factors) fundamentally limit the ability to detect resurgence from a specific group (in this example group 1 with reproduction number $R_s[1]$). Panel A shows that the grouped posterior distribution becomes less sensitive to a fixed relative change in group 1 incidence, $\Delta\lambda_{\tau(s)}[1] = \frac{i_{\tau(s)}[1] - \lambda_{\tau(s)}[1]}{\lambda_{\tau(s)}[1]}$ (the level of change increases from blue to red). Posterior distributions over $\bar{R}_s$ (the overall reproduction number across groups) are more overlapped (and tighter in variance) as $p$ rises, for fixed $R_s[1]$ (top). Panel B plots how overall resurgence detection probability $P(\bar{R}_s > 1)$ depends on the weight ($\alpha_1$, top, 0.05–1) and epidemic size ($\lambda_{\tau(s)}[1]$, middle, 20–80, $p = 2$) as well as changes in $R_s[3]$ (bottom, 0.5–1.2, $p = 3$). Decreases in $\alpha_1$ (red to blue) or $\lambda_{\tau(s)}[1]$ mean other groups mask the resurging dynamics in group 1, reducing sensitivity (curves become less steep). In the latter case the $P(\bar{R}_s > 1)$ (green with solid line at median of $\lambda_{\tau(s)}[1]$ range) is always more conservative than $P(R_s[1] > 1)$ (black with solid median line). As $R_s[3]$ falls (red to blue) the ability to detect resurgence also lags relative to that from observing group 1 (black).

distributions overlap more for a given relative change in incidence ($\Delta\lambda_{\tau(s)}[1] = \frac{i_{\tau(s)}[1] - \lambda_{\tau(s)}[1]}{\lambda_{\tau(s)}[1]}$).
Reductions in either the weight ($\alpha_1$), epidemic size ($\lambda_{\tau(s)}[1]$) or other $R_s[j \neq 1]$, further desensitise the resurgence signals i.e., decrease the gradient of the detection probability curves. This is summarised by noting that if $R_s[1] = \max_j R_s[j]$, then the sensitivity from **Eq (2)** is only matched when the resurging group dominates ($\alpha_1 \approx 1$) or if other groups have analogous $R$ i.e., $R_s[1] \approx R_s[j]$. Delays in detecting resurgence can therefore be severe. Heterogeneity on its own, however, does not force asymmetry between detecting control and resurgence.

## Discussion

Probing the performance limits of noisy biological systems has yielded important insights into the real-time estimation and control of parameters in biochemistry and neuroscience [49–51]. Although models from these fields share dynamic similarities with those in infectious disease epidemiology, there has been relatively little investigation of how real-time estimates of pathogen transmissibility, parametrised by $R$, might be fundamentally limited. This is surprising since $R$ is among key parameters considered in initiatives aiming to better systematise real-time epidemic response [41,52]. Here we explored what limits may exist on our ability to reliably detect or measure the change-points in $R$ that signify resurgence and control. By using a combination of Bayesian sensitivity analyses and minimum MSE filtering and smoothing algorithms, we discovered striking asymmetries in innate detection sensitivities. We found that, arguably, the most crucial transitions in epidemic transmissibility are possibly the most inherently difficult to detect.

Specifically, resurgence, signified by an increase in $R$ from below to above 1, can possibly be detected only 5–10 times later than an equivalent decrease in $R$ that indicated control (**Figs 2** and **3**, and **Fig A** in **S1 Appendix**). As this lag can be of the order of the mean generation time of the pathogen under study, even when case reporting is perfect and optimised detection algorithms are applied, this represents a potentially sharp bottleneck to real-time responses for highly contagious diseases. Intuition for this result came from observing that sensitivity to $R$ change-points will weaken (due to noise masking the signal) with declining epidemic sizes or case incidence, and increasing 'true' $R$, both of which likely occur in resurgent settings due to periods of subcritical spread (**Eq (2)** and **Fig 1**). The converse applies to control, which is usually enforced in larger (and less intrinsically noisy) incidence regimes. Furthermore, we found that these latencies and sensitivity issues would only exacerbate when heterogeneous groupings across geography or demography (**Eqs (4), (5)** and **Fig 4**) are considered.

An interesting corollary of these results occurs if we consider the detection of an upward shift in $R$ at large incidence. If this increase affects the majority of cases (i.e., **Eq (2)** applies), then we would detect it without significant delay because epidemic curves are now inherently less noisy. However, if incidence is large and a resurgence occurs in some subset of the cases (i.e., the upward $R$-shift is localised to group $j$ and **Eq (5)** applies) then we would still face the innate delay of a mean generation time together with further loss of sensitivity due to the cases in groups other than $j$ acting as background noise. This scenario might realistically occur when a new pathogenic variant emerges (e.g., the alpha COVID-19 variant appeared during a high incidence period in the UK [53]) or when specific age groups sustain resurgence (e.g., the 20–49 age group for COVID-19 in the USA [7]). These detection delays limit our ability to rapidly identify and target interventions at resurgent groups. Our work emphasises that the correlations among incidence, transmissibility parameters underlying this incidence and heterogeneous groups contributing to that incidence can fundamentally constrain our response sensitivity and timeliness.

Practical real-time analyses often involve grouping or data aggregation [9,13] and are subject to reporting and other latencies (e.g., if notifications, hospitalisations or deaths are used as proxies for infection incidence), which introduce additive delays on top of those we uncovered [14,54]. Consequently, we argue that while case data may provide robust signals for pinpointing when epidemics are under control (and assessing impacts of interventions), they are insufficient, on their own, to sharply resolve resurgence at low incidence. This does not devalue methods seeking to better characterise real-time $R$ changes [1,2,13,28], but instead contextualises how such inferences should be interpreted when informing policy. Given the intrinsic delays in detecting resurgences, which might associate with critical epidemiological changes such as variants of concern or shifts in population behaviours [6,7], there might be grounds for conservative policies (e.g., those of New Zealand and Australia for COVID-19 [55]) that trade off early interventions against the expense of false alarms. While the value of such policies ultimately depends on many complex economic, political and socio-behavioural factors, our study, together with works that show how lags in enacting interventions can induce drastic costs [33–36], provides a first step towards dissecting some of these trade-offs.

Moreover, our analyses suggest that designing enhanced surveillance systems, which can comprehensively engage and integrate diverse data sources [30,31] may be more important than improving models for processing case data. Fusing multiple and sometimes novel data sources, such as wastewater or cross-sectional viral loads [18,32], may present the only truly realistic means of minimizing the innate bottlenecks to resurgence detection that we have demonstrated. Approaches aimed at improving case-based inference generally correct for reporting biases or propose more robust measures of transmissibility, such as time-varying growth rates [14,41,56]. However, as our study highlights limits that persist at the gold standard of perfect case reporting and, further it is known that under such conditions growth rates and $R$ are equally informative [57], these lines of investigation are unlikely to minimise the detection limits that we have exposed.

There are three main limitations of our results. First, as we only considered renewal model epidemic descriptions with assumed generation times, which predominate real-time $R$ studies, our work necessarily neglects the often-complex contact network structures that can mediate infection spread [58] or lead to intervention-induced generation time changes [39]. However, other analyses using somewhat different approaches to ours (e.g., Hawkes processes [59]) show apparently similar sensitivity asymmetries. There is evidence that renewal models may be as accurate as network models for inferring $R$ [60], while being easier to run and fit in real time. They are also known to be equivalent to various compartmental models [61]. We do not examine the influence of generation time changes, as data on those are rarely available for routine, real-time analyses. However, as the ratio of the resurgence to control lags is 5–10, we expect this asymmetry to be robust to generation time changes, which are relatively smaller [39]. Given the flexibility of our model and that the asymmetry we discovered is contingent on low-incidence data being noisier and typical of resurgence settings, which is a model agnostic point, we expect that the intrinsic limits we have exposed are general and not model artefacts.

Second, while we analysed one common and important definition of resurgence that depends on effective reproduction numbers, other more recent definitions of epidemic re-emergence exist that are linked to complex dynamic characteristics of diseases such as critical slowing down [62]. Our aim was to understand and expose limitations of the most common surveillance data types (incidence) and the most prominent epidemic summary statistics (time-varying or effective reproduction numbers), which are among those informing policy [41], so we did not examine such metrics. Testing to see if these other characteristics also show asymmetry could be an interesting follow-up study but would require different modelling approaches. Last, we did not include any explicit economic modelling. While this is outside

the scope of this work it is important to recognise that resurgence detection threshold choices (i.e., how we decide which fluctuations in incidence are actionable) imply some judgment about the relative cost of true positives (timely resurgence detections) versus false alarms [12]. Incorporating explicit cost structures could mean that delays in detecting resurgence are acceptable. We consider this the next investigative step in our aim to probe the limits of real-time performance.

## Methods

We derive some of the mathematical formulae central to the main text. **Eq (1)** describes the renewal model [37], which simulates the spread of an epidemic, characterising how incidence at some time $s$, $I_s$, depends on the effective reproduction number at that time, $R_s$, and the total infectiousness, $\Lambda_s$. Inference under this model commonly assumes that an incidence window of size $m$ defined as $\tau(s) = \{s, s-1, \ldots, s-m+1\}$ contains all the information about $R_s$ [2]. Consequently, we have the Poisson joint log-likelihood over this window, $l_s$, (see Supplement of [40]), with grouped sums $i_{\tau(s)} = \sum_{u \in \tau(s)} I_u$ and $\lambda_{\tau(s)} = \sum_{u \in \tau(s)} \Lambda_u$, as follows.

$$l_s = \log \mathrm{P}(I_{s-m+1}^s | R_s) = i_{\tau(s)} \log R_s - R_s \lambda_{\tau(s)} + \zeta_{\tau(s)}. \tag{6}$$

In **Eq (6)**, $\zeta_{\tau(s)} = \sum_{u \in \tau(s)} - I_u! + I_u \log \Lambda_u$ is independent of $R_s$. The maximum likelihood $\tilde{R}_s$ estimate under this model solves $\frac{\partial l_s}{\partial R_s} = 0$ i.e., $\tilde{R}_s = i_{\tau(s)} \lambda_{\tau(s)}^{-1}$. The Fisher information, FI[$R_s$], defines the best achievable precision (i.e., smallest variance) around this estimate [42], and is computed from **Eq (6)** as $\mathrm{E}\left[-\frac{\partial^2 l_s}{\partial R_s^2}\right]$ [40,42]. This gives $\frac{\mathrm{E}[\sum_{u \in \tau(s)} I_u]}{R_s^2}$. Substituting $\mathrm{E}[I_u] = \Lambda_u R_s$ from **Eq (1)**, then yields the key result in the left side of **Eq (2)**.

Widely used real-time methods, such as *EpiEstim* [2] and related approaches, often calculate the posterior distribution $\mathrm{P}(R_s | I_1^s) \approx \mathrm{P}(R_s | I_{s-m+1}^s)$. This approximation is a consequence of the $m$-window assumption and is conventionally obtained by setting a conjugate gamma prior distribution i.e., $\mathrm{P}(R_s) \equiv \mathrm{Gam}(a, c^{-1})$. Hyperparameters (e.g., $a = 1$, $c = 1/5$) are often selected to ensure this prior distribution is uninformative. Applying Bayes law with the Poisson likelihood from **Eq (6)** yields $\mathrm{P}(R_s | I_{s-m+1}^s) \equiv \mathrm{Gam}(a + i_{\tau(s)}, (c + \lambda_{\tau(s)})^{-1})$.

We can compute the resurgence probability as $\mathrm{P}(R_s > 1 | I_1^s) \approx \int_1^\infty \mathrm{P}(R_s | I_{s-m+1}^s) dR_s$. This approximation also proceeds from the window-based formulation. The cumulative distribution function of the gamma posterior distribution, $\mathrm{F}_s(x)$, can be written as below for some $x$.

$$\mathrm{F}_s(x) = \mathrm{P}(R_s \leq x | I_{s-m+1}^s) = 1 - \sum_{j=0}^{a-1+i_{\tau(s)}} \frac{x^j (c + \lambda_{\tau(s)})^j}{j!} e^{-x(c + \lambda_{\tau(s)})}. \tag{7}$$

**Eq (7)** results from standard properties of gamma distributions. We compute the resurgence probability as $1 - \mathrm{F}_s(1)$, which gives the right side of **Eq (2)**. The above formulae are useful both for providing analytic insight and measuring performance of realistic estimators used in outbreak analysis, which adhere to this formulation [2,13,41,63].

These equations all feature a dependence on the choice of window size $m$. As investigated in [40] large $m$ can mean that we are slower to detect transmissibility changes, while small $m$ can lead to oversensitivity to noise. We avoid this $m$-dependence by simply using this approach to gain general, theoretical insights into detection asymmetries and latencies. Specifically, in the main text we prove that the lag in inferring resurgence is larger than that when estimating a corresponding control signal, for arbitrary window sizes (due to smaller historical incidence across suspected periods of resurgence). We then perform more detailed (but less tractable) investigations to discern the likely magnitude of these asymmetric lags.

These investigations (in **Figs 2–3** and **Fig A** in **S1 Appendix**) apply the *EpiFilter* method [15], which largely circumvents window size issues. *EpiFilter* exploits formal signal processing theory to minimise the mean squared error in the estimation of $R_s$. Its sequential predictive accuracy (i.e., it has small generalisation error) and its ability to detect change-points in real time have been verified on extensive simulations [15,28], and suggest it as a tool suitable for exploring fundamental limits on resurgence and control. This difference in methodology is signified in our notation in **Eq (3)**, which no longer uses window approximations ($\tau(s)$). There our results are direct outputs of *EpiFilter*.

Derivations for the inference equations behind the filtering and smoothing in *EpiFilter* are in [15,29]. This more general formulation allows us to go beyond the analytic insights from the *EpiEstim* type models above and limits the influence of prior distributions on results (which is particularly strong when incidence is small) since $R_s$ is a-priori uniformly distributed over some wide range ([0.01, 10] here). Consequently, we examine the problem of resurgence detection from multiple angles. The prior distributions used in all methods have mean and median above 1 so that any delays we find in detecting resurgence are the minimum possible.

The trends uncovered in **Eq (4)**, where heterogeneity or grouping is explored, are within the *EpiEstim* framework, but will be valid for *EpiFilter* and general *R*-estimation methods, since they result from the properties of convex sums and averages only. Last, while our conclusions may appear limited due to their dependence on renewal models, we note that renewal models (i) can describe realistic transmission patterns for many diseases with accuracies comparable to that of more detailed network-based models [60] (ii) are the dominant model for measuring real-time outbreak changes [1,41,60] and (iii) are able to equivalently represent the dynamics of prevailing compartmental models, such as the SEIR model, depending on the form of the generation time distribution considered [61].

## Supporting information

**S1 Appendix. This provides additional Figs A-C. Fig A**: Resurgence and control dynamics of COVID-19. We repeat the simulations from **Fig 2** but for realisations of COVID-19 epidemics. **Fig B:** Incidence curves for COVID-19. We present the simulated counts of daily new cases that underlie the results of **Fig A**. **Fig C:** Incidence curves for Ebola virus disease. We present the counts of daily new cases that underlie the results of **Fig 2** of the main text. (PDF)

## Author Contributions

**Conceptualization:** Kris V. Parag.

**Formal analysis:** Kris V. Parag.

**Funding acquisition:** Kris V. Parag, Christl A. Donnelly.

**Investigation:** Kris V. Parag.

**Methodology:** Kris V. Parag.

**Project administration:** Kris V. Parag.

**Software:** Kris V. Parag.

**Validation:** Christl A. Donnelly.

**Writing – original draft:** Kris V. Parag.

**Writing – review & editing:** Kris V. Parag, Christl A. Donnelly.

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
