## [Decision Letter · Decision Letter 0]

4 Feb 2022

Dear Dr Parag,

Thank you very much for submitting your manuscript "Fundamental limits on inferring epidemic resurgence in real time using effective reproduction numbers" for consideration at PLOS Computational Biology. As with all papers reviewed by the journal, your manuscript was reviewed by members of the editorial board and by several independent reviewers. The reviewers appreciated the attention to an important topic. Based on the reviews, we are likely to accept this manuscript for publication, providing that you modify the manuscript according to the review recommendations.

Sincerely,

Joseph T. Wu

Associate Editor

PLOS Computational Biology

Virginia Pitzer

Deputy Editor-in-Chief

PLOS Computational Biology

[LINK]

Reviewer's Responses to Questions

**Comments to the Authors:**

Reviewer #1: In "Fundamental limits on inferring epidemic resurgence in real time using effective reproduction numbers", Parag and Donnelly examine out ability to detect changes in epidemic growth, as characterized by the reproductive number R, in real time. They find that there are delays in our ability to detect changes in the reproductive number, particularly changes around the critical threshold of R being above or below 1, that are inherent in the generating process and hence cannot be overcome by improved analysis of epidemic curves. In many ways there results are obvious, shifts in R cannot be detected without data, and since that data comes from subsequent generations of infection that are necessarily delayed by the generation time of the disease, there is an upper limit to how quickly we can detect these shifts. Likewise, as more cases are present when epidemics shift to a decline than when they shift to a resurgence, statistical theory tells us the former will be easier to detect than the latter.

My statement that these conclusions are in retrospect obvious is in no way a negative comment on this work…much of the best science seems obvious in retrospect. The thorough and comprehensive way in which the authors prove their point and quantify many of these delays is compelling and invaluable, and I found no technical problems with their work. I found the use of classical statistical measures, such as Fisher information, to characterize many of the results a particularly compelling. Hence, I think this analysis is a strong and important contribution to the literature.

As detailed below, I do think the presentation (particularly the figures) could be made more clear and that the authors could be a little bit more nuanced in their discussion of the practical impact of these results. However, these are simply improvements to what is already and interesting and informative paper.

Specific Comments:

1 Abstract, "This belies epidemic…"

I don't think this means quite what the authors intend, or I am misunderstanding the rest of the sentence.

2 Abstract, "Responses to recrudescent…"

I think this misses the nuance that is around a similar statement in the main text. I.e., it does not come across that what is being argued is that improvements to analysis of epidemics curves are less important than improving surveillance.

3 Introduction, "…enhancing syndromic…Such systems…"

I feel like the argument here is broader than syndromic surveillance, as highlighted by the reference to wastewater surveillance later. I.e., the authors appear to be arguing that novel methods beyond syndromic surveillance might be useful.

4 Figures 1-4

I will admit to finding the figures a bit hard to decode, and wonder if something can be done to make the meaning of the colors, axes, etc. a bit more clear.

5 Discussion, "…there are grounds….that enact interventions…at the expense of false alarms"

I wonder if this argument goes too far beyond what the analysis in the paper supports, and might warrant a more nuanced discussion. Without an analysis of the cost of false alarms, it is hard to say if a conservative approach is warranted or not. Further, this decision will depend on the cost of an intervention over the short and long term, which will likely be specific to intervention and context. As to the latter point, my perception is that in the United States public health agencies had the political capital to implement interventions only a limited number of times, while in some other countries there was more acceptance or repeated imposition of control measures during the COVID-19 pandemic. I certainly don't think it is incumbent on the authors to wade into all such thorny issues, but I think it would be worthwhile for them to acknowledge they exist and that their analysis represents only a first step in figuring out how to better optimize responses. [I now see that this was addressed a bit in the last paragraph, but still think it warrants a bit more discussion here].

6 Discussion, "…enhancing syndromic surveillance…."

See above comments about limiting to syndromic.

Reviewer #2: Please see attachment.

**Have the authors made all data and (if applicable) computational code underlying the findings in their manuscript fully available?**

Reviewer #1: Yes

Reviewer #2: Yes

PLOS authors have the option to publish the peer review history of their article (what does this mean?). If published, this will include your full peer review and any attached files.

Reviewer #1: No

Reviewer #2: No

Figure Files:

Data Requirements:

Reproducibility:

References:

---

## [Decision Letter · Decision Letter 1]

8 Mar 2022

Dear Dr Parag,

We are pleased to inform you that your manuscript 'Fundamental limits on inferring epidemic resurgence in real time using effective reproduction numbers' has been provisionally accepted for publication in PLOS Computational Biology.

Best regards,

Joseph T. Wu

Associate Editor

PLOS Computational Biology

Virginia Pitzer

Deputy Editor-in-Chief

PLOS Computational Biology

Reviewer's Responses to Questions

**Comments to the Authors:**

Reviewer #1: The authors have done a good job of addressing most of my concerns, and I think this paper will be a valuable contribution to the literature. I do still find the figures a touch challenging, but understand that the concepts being conveyed are complex.

Reviewer #2: Thank you for your comprehensive response to the points raised.

**Have the authors made all data and (if applicable) computational code underlying the findings in their manuscript fully available?**

Reviewer #1: Yes

Reviewer #2: Yes

PLOS authors have the option to publish the peer review history of their article (what does this mean?). If published, this will include your full peer review and any attached files.

Reviewer #1: No

Reviewer #2: No

---

## [Editor Report · Acceptance letter]

6 Apr 2022

PCOMPBIOL-D-21-02218R1 

Fundamental limits on inferring epidemic resurgence in real time using effective reproduction numbers

Dear Dr Parag,

I am pleased to inform you that your manuscript has been formally accepted for publication in PLOS Computational Biology. Your manuscript is now with our production department and you will be notified of the publication date in due course.

With kind regards,

Livia Horvath
